# Effect of Phenol Formaldehyde Resin Penetration on the Quasi-Static and Dynamic Mechanics of Wood Cell Walls Using Nanoindentation

**DOI:** 10.3390/nano9101409

**Published:** 2019-10-02

**Authors:** Xinzhou Wang, Xuanzong Chen, Xuqin Xie, Zhurun Yuan, Shaoxiang Cai, Yanjun Li

**Affiliations:** 1College of Materials Science and Engineering, Nanjing Forestry University, Nanjing 210037, China; xzwang@njfu.edu.cn (X.W.); zystcxz@163.com (X.C.); yuanzr0311@163.com (Z.Y.); caishaoxiang@126.com (S.C.); 2Dehua Tubao New Decoration Material Co., Ltd., Huzhou 313200, China; xiexuqinvip@163.com

**Keywords:** phenol formaldehyde resin, wood modification, cell-wall mechanics, nanoindentation, modulus mapping

## Abstract

To evaluate the effects of phenol formaldehyde (PF) resin modification on wood cell walls, Masson pine (*Pinus massoniana* Lamb.) wood was impregnated with PF resin at the concentrations of 15%, 20%, 25%, and 30%, respectively. The penetration degree of PF resin into wood tracheids was quantitatively determined using confocal laser scanning microscopy (CLSM). The micromechanical properties of the control and PF-modified wood cell walls were then analyzed by the method of quasi-static nanoindentation and dynamic modulus mapping techniques. Results indicated that PF resin with low molecular weight can penetrate deeply into the wood tissues and even into the cell walls. However, the penetration degree decreased accompanying with the increase of penetration depth in wood. Both the quasi-static and dynamic mechanics of wood cell walls increased significantly after modification by the PF resin at the concentration less than 20%. The cell-wall mechanics maintained stable and even decreased as the resin concentration was increased above 20%, resulting from the increasing bulking effects such as the decreased crystallinity degree of cellulose. Furthermore, the mechanics of cell walls in the inner layer was lower than that in the outer layer of PF-modified wood.

## 1. Introduction

Phenol formaldehyde (PF) is one of the most common resins used in the wood industry, such as in the manufacturing of oriented strandboard (OSB), plywood, and cross-laminated timber (CLT), due to the advantages of low initial viscosity, water repellency, and excellent temperature stability [1,2,3]. In addition, the water-soluble PF has been widely applied as a modifier for improving the properties of wood, such as its strength, dimensional stability, and biological durability [4,5,6,7,8]. The impregnation of PF oligomers into wood also provides persistent protection against biological and weathering for in-service wood due to the higher resistance leachability of thermosetting PF adhesive. PF impregnation has been regarded as one feasible technique for wood modification because it is less toxic to the environment [9,10,11]. This technique is also facing some problems during its industrial application, such as the low modification efficiency accompanying with the high dosage of PF adhesive. In the past decades, most work has been paid on the contributions of the PF filling in the wood cell lumen on the modification efficiency while ignoring the contributions of PF penetration in wood cell wall.

A good cell wall penetration plays an important role on wood modification effects. Thus, more researchers have focused on investigating the penetration phenomena of PF resin into wood for clarifying the modification mechanisms. The penetration phenomena at the nanometer, micrometer, or larger scale through wood cell lumens, interconnecting pits of the wood cells, and the nanovoids with cell walls has been observed by optical microscopy, electron microscopy, and X-ray fluorescence microscopy in the last few decades [12,13,14]. In particular, the nano-penetration of PF resin in wood cell walls has been intensively analyzed because it can provide the opportunity for cross-linking reactions between PF resins and the functional groups in wood polymers. In the previous work, both the chemical and mechanical properties of wood cell walls penetrated by PF resin during the wood gluing were characterized by AFM-based infrared spectroscopy (AFM-IR), which further confirmed that the physical filling and the possible chemical reactions are increasingly considered as the major contributions to the improved wood properties such as bending strength and hardness [15,16]. It is well-known that the depth and capacity of penetration was prone to be influenced by the compatibility between the chemical and physical characteristics of the wood and resin and by the process used [12]. However, only limited attempts have been made to analyze the chemical, physical, and mechanical properties of the wood cell wall after PF resin modification by vacuum-pressure impregnation. In particular, for PF-modified wood, it is still unclear whether the penetration degree affects the properties of wood cell wall. To this purpose, methods with adequate spatial resolution are necessary.

Nanoindentation (NI) is an effective method for measuring the mechanical properties of various materials at the micro-level, which has been widely introduced on plant cell walls and polymer films [17,18,19,20]. However, conventional NI cannot adequately characterize the viscoelastic behavior of wood cell walls, especially after thermosetting PF resin impregnation. The modified wood is usually used as a structural material which is often exposed to the oscillation stress in practical use. Thus, the capacity of storing and dissipating energy of wood is quite closely related to the in-service performance [21]. Dynamic modulus mapping technique is an alternative approach to acquire the continuous contact stiffness with finer spatial granularity through scanning probe [22,23]. A modulus mapping was generated to provide dynamic mechanical properties, including the storage modulus (*E*r′) and loss modulus (*E*r″). Recently, this method has been successfully applied for analyzing the dynamic mechanical properties of wood cell walls. The results indicated that the modulus mapping may be an ideal method for biocomposite [24,25].

In this study, the in situ quasi-static and dynamic mechanical properties of wood tracheid cell wall modified by phenol formaldehyde resin was investigated using nanoindentation and dynamic modulus mapping technique. In particular, the local variation in micromechanics of PF-modified cell walls at different PF concentrations was analyzed to increase the understanding of wood chemical modification mechanism.

## 2. Materials and Methods

### 2.1. Materials

The wood species chosen was Masson pine (*Pinus massoniana* Lamb.), which is one of the most widely distributed tree species in the subtropical regions of China. The 40-year-old wood was collected from a plantation located in Fujian Province, China. Phenol formaldehyde (PF) resin (Dynea Co., Ltd., Zhaoqing, China) with the solid contents of 48% and the viscosity of 150 mPa·s at 25 °C were used for wood modification.

### 2.2. Modification Process

Sapwood blocks with the dimensions of 20 × 20 × 100 mm (tangential × radial × longitudinal) were cut from the tree trunk at same height and around 18th to 24th growth ring (the average width of the ring is about 3 mm). Wood blocks were conditioned at 65 ± 3% relative humidity (RH) and 20 ± 2 °C until they reached the moisture content (MC) of ≈ 11%. The average density of wood blocks after conditioning was 0.49 g·cm^‒3^. PF resin was diluted by distilled water into new resin concentrations of 15%, 20%, 25%, and 30% for separate treatment, respectively. As shown in Figure 1, 15 replicate oven-dried samples in each treatment group were soaked in resin solution at a chamber under the vacuum of 0.09 MPa for 30 min and at the pressure of 0.8 MPa for 2 h. The modified wood samples were air-dried at room temperature and were then cured at 130 °C for 2 h in an oven.

### 2.3. MALDI-TOF Analysis

AB-5800 MALDI-TOF instrument (AB SCIEX, Framingham, MA USA) was used to analysis the molecular weight distribution of PF resin. The irradiation source was a pulsed nitrogen laser with a wavelength of 337 nm. The duration of a single laser pulse was 3 ns. The measurements were conducted under the following conditions: Polarity-positive, flight path-linear and 25 kV acceleration voltage. Before curing, PF resin samples were dissolved in acetone (5 mg/mL). The sample solutions were mixed in the matrix of 2, 5-dihydroxy benzoic acid (DHB). For the enhancement of ion formation, 0.1 M NaCl was added to the matrix. About 1 μL of the resulting solution were placed on the MALDI target. After evaporation of the solvent, the MALDI target was introduced into the spectrometer.

### 2.4. Confocal Laser Scanning Microscopy Analysis

The microscopic distribution of PF resin within the wood was examined using a LSM 710 inverted confocal microscope (Carl Zeiss, Oberkochen, Germany) under the fluorescence mode at the excitation laser wavelength of 488 nm and 550 nm for the identification of PF resin and wood components, respectively, which had two emission wavelengths of 450–550 nm and 500–550 nm. The cross-sectional slices with the thickness of 20 μm were prepared from the outer and inner layers of wood samples (Figure 1) and then mounted by Glycerol for acquiring images at 40 magnifications. In order to evaluate the penetration degree of PF resin in wood samples, the area measurement and calculation of PF resin area penetrated within wood tracheids at different resin concentrations was carried out with the aid of image processing software ZEN (Carl Zeiss, Oberkochen, Germany) and Equation (1):(1)$$Relative\;penetration\;degree\;\left(\%\right)=\frac{total\;area\;of\;PF\;resin\;penetrated\;in\;tracheids}{total\;area\;of\;wood\;section}$$

### 2.5. XRD Measurement

The control and modified wood flour were prepared for XRD analysis. The crystalline structures were analyzed by Uitima IV X-ray diffractometer (Rigaku Inc., Tokyo, Japan) using CuKα radiation with a scanning speed of 4°·min^−1^ at diffractograms range of 4–50° (2θ). The relative crystallinity degree (*C*_r_*I*) was estimated based on the method proposed by Segal et al. (1959) [26]:(2)$$C_{r}I\left(\%\right)\;=100\;\times \;\frac{\left(I_{002}\;-\;I_{\mathrm{am}}\right)}{I_{002}}$$
where *I*_002_ and *I*_am_ represent the intensity of the 200 crystalline peak and diffraction of the amorphous part, respectively.

### 2.6. FTIR Measurement

The chemical functional groups of the control and modified wood samples were analyzed using a VERTEX 80V FTIR spectrometer (Bruker Corporation, Karlsruhe, Germany). Wood samples was first ground and screened into flour (40 and 60 mesh). All spectra were collected in the range 1800–800 cm^−1^ with a resolution of 4 cm^−1^.

### 2.7. Quasi-Static Nanoindentation

Smaller wood samples were cut from the outer and inner layers in modified wood blocks and their transections were polished by ultra-microtome (Leica EM UC7, Wetzlar, Germany) for nanoindentation (NI). As shown in Figure 2, the mechanics of the secondary cell wall of control and modified wood tracheids were tested by Hysitron TriboIndenter system (Hysitron Inc., Minneapolis, MN, USA) in a quasi-static loading model: Loading, holding at the peak load of 400 μN, and unloading for 5 s, respectively. The locations for NI were selected by scanning probe micrographs taken with the indenter tip (Figure 2b). Before indenting, the specimens were conditioned at ambient temperature of 21 °C and relative humidity of 55% for 36 h. About 30 indents placed in the valid positions (without apparent defects, edge of cell wall, etc.) were analyzed. The reduced elastic modulus (*E*_r_) and hardness (*H*) were calculated from the load-depth curves according to Oliver and Pharr’s method (1992) [27]:(3)$$H=\frac{P_{\max}}{A}$$
where *P*_max_ is the peak load, and *A* is the projected contact area of the tips at peak load.
(4)$$E_{r}=\frac{\sqrt{\pi }}{2\beta }\frac{S}{\sqrt{A}}$$
where *E*_r_ is the combined elastic modulus; *S* is initial unloading stiffness; and *β* is a correction factor correlated to indenter geometry (*β* = 1.034).

### 2.8. Dynamic Modulus Mapping

Dynamic modulus mapping was operated on the same Triboindenter equipped with a nano-DMA model transducer. A small sinusoidal force of 6 μN at a frequency of 200 Hz was superimposed on top of a constant quasi-static force of 12 μN during the raster scanning imaging process. In this experiment, raster scanning was performed over a 15 × 15μm^2^ region covering both the PF resin filled in cell lumen and wood cell wall. As the tip scanned across the sample surface, the amplitude of displacement was maintained at 0–1.5 nm and the phase of the resulting transducer displacement signal was measured by a lock-in amplifier. The phase information was used to determine the local indentation modulus of the sample at each pixel of the imaging process. A modulus map containing the dynamic modulus information was collected at a pixel resolution of 256 × 256. The storage and loss modulus values were computed on the basis of the measured stiffness at each pixel according to the method of Asif et al. [28,29].

## 3. Result and Discussions

### 3.1. Molecular Weight of PF Resin

Figure 3 shows the MALDI-TOF spectra of PF resin before curing. The spectra showed clearly repetitive patterns of peaks that allow the identification of specific oligomer series present in the PF resin. The peaks in the spectra represented PF compounds + Na^+^ because NaCl was used as an ionizing agent in this experiment. For instance, the peak at 313 Da was calculated as 290 Da (molar mass MW of PF dimers) + 23 Da (MW of Na^+^). The major peak-to-peak mass increments observed were 106 Da and 136 Da corresponded to repeating units A and B in theory as shown in Figure 3. The PF resin used in this study was mainly composed of oligomers up to 2–4 phenolic nuclei, which were mainly bonded by methylene bridges [30]. As described in the MALDI-TOF spectra, the molecular weight of most oligomers in the uncured PF resin was less than 500 Da. The low molecular weight may be beneficial for the penetration of PF resin into the wood and even into cell walls [15].

### 3.2. Penetration of PF Resin in Wood

Figure 4 shows the penetration of PF resin with different concentrations in different layers of wood. The green light spot in the CLSM microphotographs clearly shows that the PF resin has trapped in the lumens of many tracheids, indicating that the PF resin has penetrated deeply into the wood tissues under the action of exterior pressure. It can be observed from Figure 4 that more and more PF resin penetrated into wood samples with the increasing resin concentration. However, the penetration area of PF resin in both outer and inner layers increased slowly when the concentration was above 20%. This finding was in agreement with the previous study that the PF resin with higher concentration could decrease the permeability of resin in wood and the weight percent gain increased slowly when the concentration was above 20% [31]. As expected, the resin penetration area of outer layer was larger than that of inner layer. During the impregnation process, the chemicals contacted with the outer layer primarily, entered the interior of the wood through the wood tracheids, and then circulated through pits in axial and transverse direction [32,33]. The increased viscosity of the resin solution resulted from the water loss during this period may retard the penetration in the inner layer of wood.

In order to quantitatively assess the impregnation effectiveness, the penetration area of PF resin (green color) was measured by image processing software and then the relative penetration degree (RPG) was calculated according to Equation (1). As shown in Figure 5, in agreement with CLSM microphotographs, the RPG initially increased obviously at a concentration ranged from 20% to 25% and then remained stable with an increase in concentration. The RPG of the outer layer in wood treated with 25% PF resin reached to the maximum of 22.4%, while the largest RPG of the inner layer was 5.2% when the resin concentration was 20%.

### 3.3. Crystallinity Analysis

Figure 6 shows the X-ray diffractograms of the control and modified wood samples. As presented in the diffractograms, three types of cellulose patterns were identified, with 110, 200, and 040 peaks observed around 14.8°, 22.2°, and 34.5° 2θ angles, respectively [34]. The most significant diffraction peak (200) was at 2θ = 22.163° for the control, whereas the diffraction peaks slightly shift to a low angle after PF resin modification, which can be explained by macroscopic residual stress and lattice distortion ascribed to uneven diffusion of chemicals inside wood [35]. The relative crystallinity (*C*_r_*I*), defined as the percentage of cellulose crystallization area in the whole cellulose, was closely related to the physical and mechanical properties of wood [36]. Thus, the relative crystallinity was then calculated according to Equation (2). The *C*_r_*I* of wood was reduced with the addition of PF resin. When the sample was treated with 20% PF resin, the *C*_r_*I* was 47.72%, which was 6.38% lower than that of the control. This result can be attributed to the fact that the chemical reactions between -OH groups mainly existed in the amorphous region of the wood and PF resin may increase the mass of the amorphous region. Meanwhile, the crystallization region was stable [33].

### 3.4. FTIR Spectrum Analysis

FTIR was applied to examine possible interactions between the wood cell-wall materials and PF adhesive. Figure 7 displays the typical raw spectra for the oven-dried wood samples after eight weeks. For the control wood, the prominent band at 1045, 1269, and 1736 cm^−1^ arise from the C-O stretching vibration and the C = O stretching vibration in acetyl groups in cellulose and hemicellulose, respectively. The characteristic bands at 1609, 1464, and 1269 cm^−1^ originate from the C = C stretching vibration in the aromatic ring of the phenol, C-O stretching vibration, and the C-H methylene bridge in PF adhesive, respectively [37]. As shown in Figure 7, the increased intensity at 1609, 1464, and 1269 cm^−1^, which can be attributed to the penetration of PF resin. Moreover, a relatively broader band between 1150 and 1000 cm^−1^ appeared in the spectra of modified wood, corresponding to the asymmetric stretching vibration of C-O-C aliphatic ether, which confirmed the chemical reactions of the -OH groups of wood and PF adhesive [16,38]. The O–H stretching of hydroxyls around 3414 cm^−1^ of the wood modified by 20% PF resin becomes a little narrower compared to the control, which may be attributed to the interactions between PF resin and wood [39]. However, the band of O–H stretching becomes broader with the increase of PF concentrations, which may be ascribed to the increase in the amount of self-curing PF resin.

### 3.5. Quasi-Static Mechanics of Wood Cell Wall

Figure 8 displays the elastic modulus (*E*_r_) and hardness (*H*) of the control and modified wood cell walls in outer and inner layers. As compared to the control, both the *E*_r_ and *H* values increased significantly after modification by the PF resin with the concentration less than 20%. For example, the *E*_r_ and *H* values of the wood cell walls modified with 20% PF resin increased by about 17.7% and 35.5%, which further confirmed that some PF molecules has penetrated into the cell wall. The cross-linking between -OH groups of wood polymer with and resins may reinforce the cell wall [14]. The mechanics of inner layer of modified wood were lower than that of outer layer and were stable when the resin concentration was above 20%, resulting from the lower penetration degree of PF resin with higher concentration into wood cell wall. However, the cell-wall mechanics of outer layer in the wood modified by PF resin with the concentration above 20% decreased with the increasing resin concentration, which can be attributed to the increasing bulking effects after deposition of lots of resin in the cell walls. For instance, the decreased crystallinity index of cellulose may have also contributed to this effect. In addition, the results in our previous work confirmed that the increased mechanics of wood cell walls induced by PF penetration also positively affect the physical and mechanical properties of wood on the macro-scale. The antiswelling efficiency (ASE) and antishrinking efficiency (ASE’) of the wood treated with 20% PF resin reached 54% and 50%, and the hardness of the transverse section and the compressive strength parallel to the grain of the wood increased by about 34% and 11%, respectively, compared to the control. [31].

### 3.6. Dynamical Mechanics of Wood Cell Wall

Figure 9a presents the contact force image of the modified wood cell wall, computed from the measured displacement and mechanical quantity transducer correlated with the position during scanning. The corresponding modulus maps were obtained from DMA mode measurements. Both the storage modulus and loss modulus were recorded as a map in Figure 9b,c. As displayed in the map, quantitative variations in modulus values were shown in different colors. It can be observed from the scale bar that there is a large variation on the storage modulus among PF resin filled in the lumen, cell wall, and middle lamella. In contrast, the variation on the loss modulus is much smaller and the values were also smaller than storage modulus.

After modulus mapping, 256 lines of data were available with 256 data points in each line. These points in one line can be transferred to modulus data for analyzing the variations in different materials with high resolution. As show in Figure 9b,c, one line across the PF resin filled in the lumen, cell wall, and cell middle lamella (CML) was chosen to obtain the modulus data and plot the profile of storage modulus and loss modulus. As shown in Figure 10, the cell walls exhibited greater modulus than the CML and the PF resin filled in cell lumen. It is also interesting that the modulus across the thickness of the cell walls clearly illustrated the gradient distribution across the S_1_ to S_3_ layers. The modulus of S_2_ layer was obviously higher than that of S_1_ and S_3_ layers, which was in accordance with the general view that the S_2_ made the major contribution to the wood mechanics due to its small microfibril angle and higher cellulose content [40,41]. In addition, it is evident that the storage modulus was comparable to the reduced elastic modulus obtained by means of quasi-static nanoindentation, indicating that the nano-DMA testing is preferable for studying wood materials.

Based on the method of modulus mapping, the teen replicate area of the control and modified wood sample was scanned by nano-DMA. The average storage modulus (*E*′) and loss modulus (*E*″) of secondary cell wall (S_2_) from the outer and inner layers were calculated and presented in Figure 11. The initial *E*′ and *E*″ values of the secondary cell walls of the control were 21.99 GPa and 4.22 GPa, respectively. After modification, the *E*′ gradually increased first and then remained stable or even decreased with an increase in resin concentration, reaching the maximum of 28.71 GPa and 26.85 GPa with a concentration of 20% and 15% for the outer and inner layer, respectively. The higher *E*′ of cell wall may be due to the inhibitions of the main chain movements after being penetrated by PF monomers [42,43]. Except for the physical filling of PF monomers in the nanovoids within the cell walls, the interconnected networks formed within the microfibrils by chemical reaction are another influence factor [44]. The loss modulus (*E*″) of both the control and modified wood cell walls was much smaller than the storage modulus (*E*′). Moreover, there was no obvious change in the loss modulus (*E*″) of wood after modification.

## 4. Conclusions

The molecular weight of most oligomers in the uncured PF resin was less than 500 Da, which was beneficial for the penetration of PF resin into the wood and even into the cell walls. The physical filling of PF monomers in the nanovoids within wood cell walls and the interconnected networks formed by chemical reaction made a positive contribution on the mechanical properties of wood. Both the quasi-static and dynamic mechanics, including the elastic modulus (*E*_r_), hardness (*H*), and storage modulus (*E*″) of cell walls, were improved obviously after modification by PF resin at a low concentration. However, as the resin concentration was above 20%, the cell-wall mechanics became stable and even slightly decreased due to the increasing bulking effects. Furthermore, the cell walls in the outer layer of modified wood exhibited greater modulus than that of the inner layer ascribed to the higher penetration degree of PF resin.

## Figures and Tables

**Figure 1 nanomaterials-09-01409-f001:**
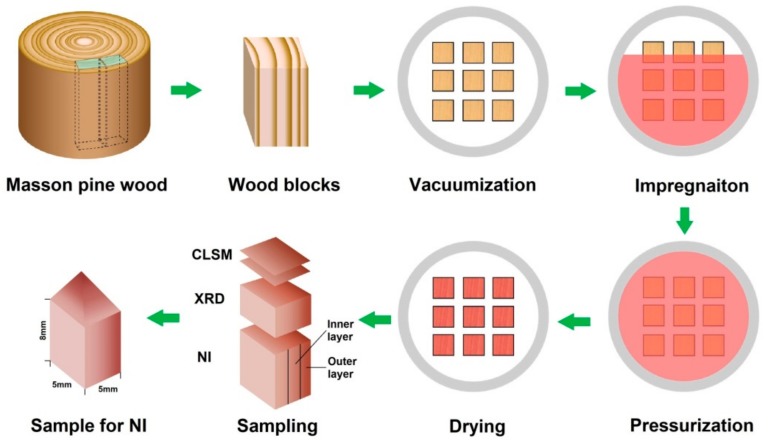
Schematic diagram of wood impregnation and sample preparation.

**Figure 2 nanomaterials-09-01409-f002:**
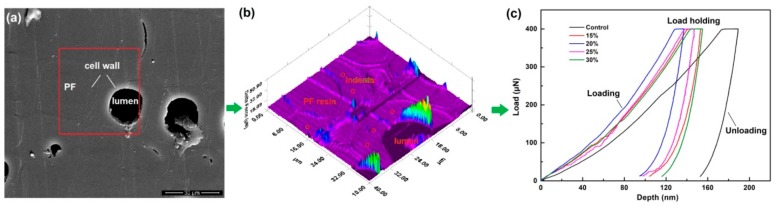
Wood sample preparation for nanoindentation test: (**a**) Microscope images showing the test location; (**b**) Scanning probe microscopy (SPM) image showing the positioning of indents; (**c**) Typical indentation load-depth curves of cell walls.

**Figure 3 nanomaterials-09-01409-f003:**
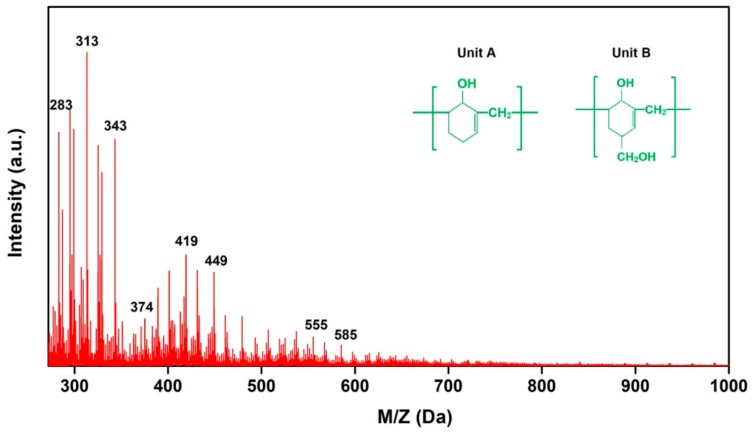
The MALDI-TOF spectra of phenol formaldehyde (PF) resin.

**Figure 4 nanomaterials-09-01409-f004:**
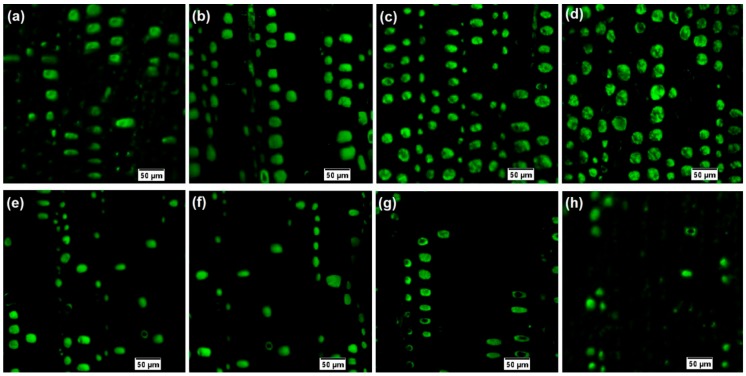
Typical confocal laser scanning microscopy (CLSM) microphotographs of the cured PF resin trapped in the wood tracheids: (**a**–**d**) PF resin with the concentrations of 15%, 20%, 25%, and 30% in the outer layer of wood, respectively. (**e**–**h**) PF resin with the concentrations of 15%, 20%, 25%, and 30% in the inner layer of wood, respectively. Cured PF resins in the lumen of tracheids are showed in green color.

**Figure 5 nanomaterials-09-01409-f005:**
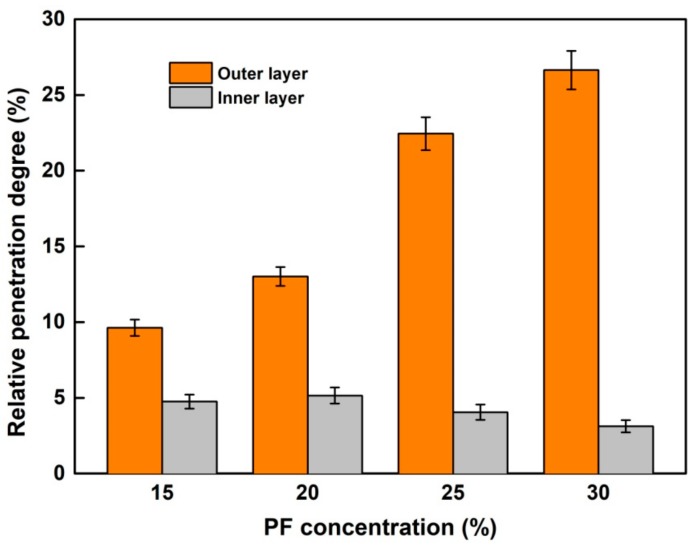
The relative penetration degree (RPG) of PF resin in the wood tracheid lumens.

**Figure 6 nanomaterials-09-01409-f006:**
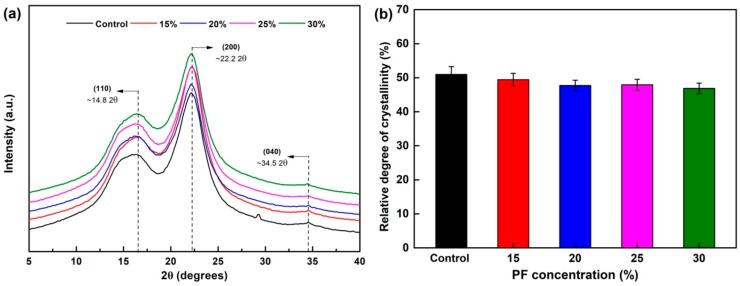
XRD analysis of the control and PF-modified wood: (**a**) XRD spectra and (**b**) the degree of crystallinity.

**Figure 7 nanomaterials-09-01409-f007:**
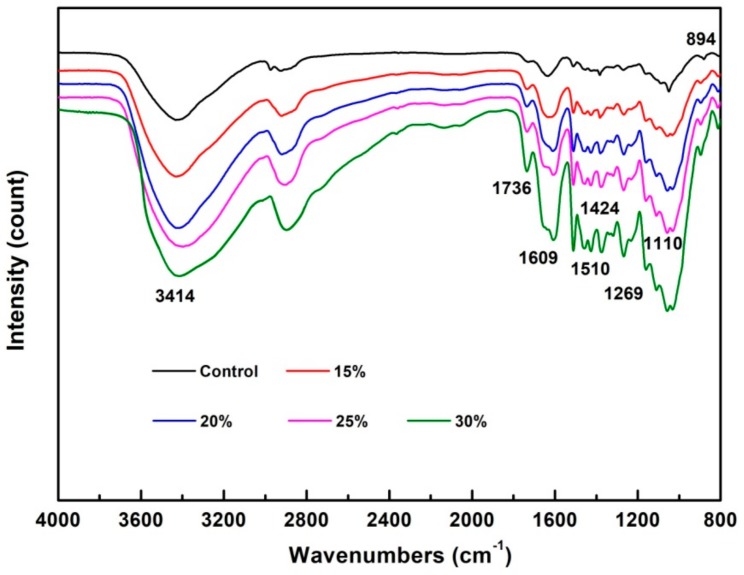
FTIR spectra of the control and PF modified wood.

**Figure 8 nanomaterials-09-01409-f008:**
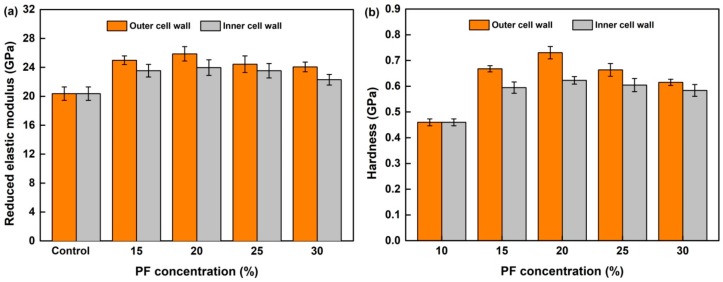
Mechanical properties for the control and PF-modified wood cell walls. (**a**) Reduced elastic modulus and (**b**) hardness.

**Figure 9 nanomaterials-09-01409-f009:**
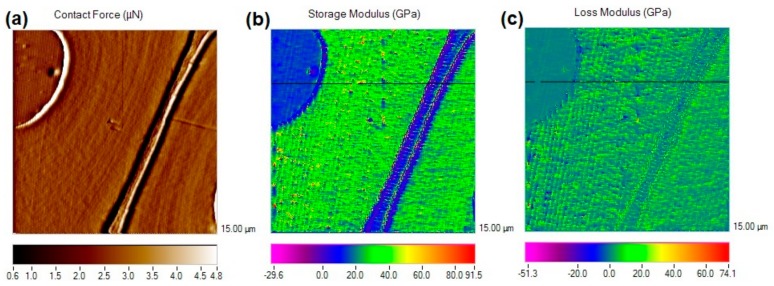
Modulus mapping of PF-modified wood tracheids: (**a**) Contact force image; (**b**) Storage modulus image; (**c**) Loss modulus image.

**Figure 10 nanomaterials-09-01409-f010:**
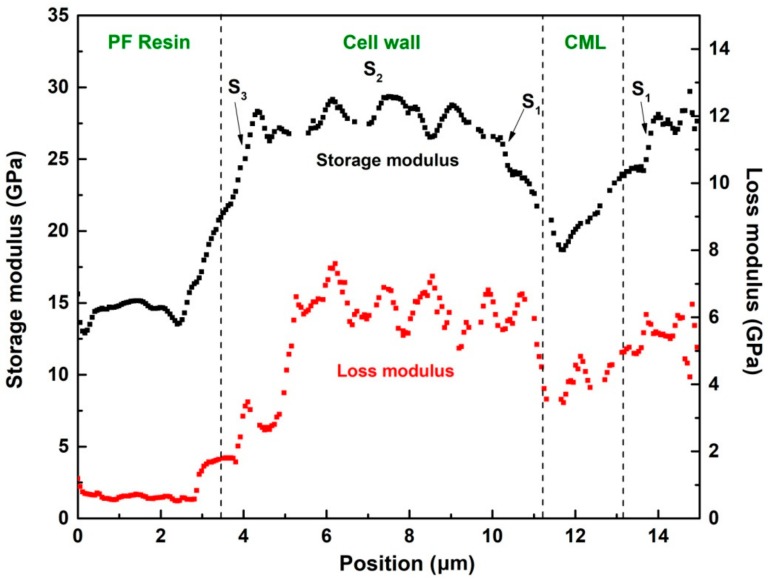
Profiles of storage modulus and loss modulus of PF-modified wood tracheids.

**Figure 11 nanomaterials-09-01409-f011:**
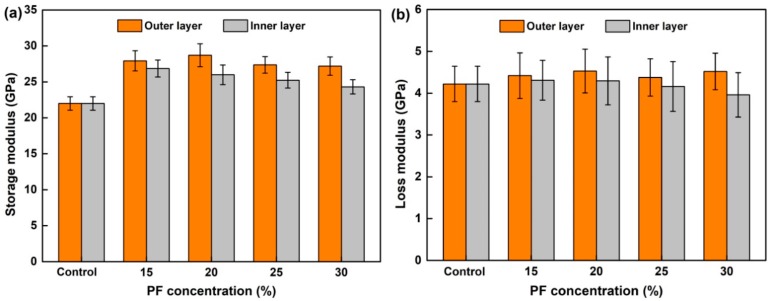
Dynamic mechanical properties of wood tracheids: (**a**) Storage modulus and (**b**) loss modulus.

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
