# Peer review of "Effect of Phenol Formaldehyde Resin Penetration on the Quasi-Static and Dynamic Mechanics of Wood Cell Walls Using Nanoindentation"

_nanomaterials, 2019, doi:10.3390/nano9101409_

Round 1

Reviewer 1 Report

This manuscript described the modification of wood with different concentration of a phenol formaldehyde (PF) resin, using different characterization techniques. Results are in good agreement with the different techniques used, however some extra characterization experiments are missed, such as:

The mechanical properties of the wood studied in this manuscript were exclusively viscoelastics properties, such as static and dynamic storage and loss moduli. However, it would be interesting to know the shear and compression strength improvement and bending or stiffness alteration after wood modification.

The phenol formaldehyde resin reacts with free OH- groups of the wood, as authors mentioned in the manuscript. The crosslinking reaction between the resin and the OH- groups was not studied or yield quantified, so maybe the chemical reaction could continue with time due to the existence of free react groups inside the modified materials. Then, it will be convenient to repeat the characterization techniques after 2 or 3 weeks and compare all the mechanical properties results with the initial ones.  

Authors do not specify the applications related to this modified wood. It is well known that wood modification with PF usually modifies the color, odour and water resistance properties. If this wood would be used for furniture, for example, these properties are quite important to be studied aswell.

The viscoelastic properties were improved noticeably in the outer layer of modified wood, but the mechanics of inner layer were lower. Have the authors done extra studies to improve the wood impregnation method to get more homogeneous mechanic properties? The ideal modified wood would have the same mechanical properties in the whole material surface.

Author Response

Dear Reviewer, Thank you so much for spending much time to give us lots of good comments on our manuscript and giving us the opportunity to revise our manuscript. We have carefully taken the comments into consideration in preparing our revision.

Reviewer 2 Report

In my opinion it is an interesting article about the effect of resin penetration on the mechanics of wood cell walls. It can be printed without changes.

There are only two typing errors which should be corrected:
- page 4 / line 152: the link 'Figure 1' is wrong, must be 'Figure 3'
- page 9 / line 273: in graph (a) of Fig. 10 the Legend is incomplete:
                            'Outlayer'  --> 'Outer Layer' & 'Inner' --> 'Inner Layer'

Author Response

Dear Reviewer,

Thank you so much for spending much time to give us lots of good comments on our manuscript. We have carefully taken the comments into consideration in preparing our revision.

Reviewer 3 Report

Introduction,

This is interesting topic. Wood preservation has faced a lot of issues recently. The majority of the environmentally challenging solutions have been removed from the market or banned. Tropical wood species are less and less desired. So new modification technologies have been developed. Modification of wood with PF resins is old technology, that has been somehow reintroduced in he recent years. This is still very niche technology. One of the issues related with this technology, that are not being discussed in present paper are formaldehyde emissions in production and use phase. In addition, end of life issues, utilization of fossil sources for PF production, does not make this technology green. I suggest the authors to fully consider pro and contra arguments in order to make respective introduction trustworthy.

In addition, I am not sure If respective paper fits within category nanomaterials. I do agree that some of the analytical methods used utilze technology on the nanoscale but overall this material is not fully nano …

Line 34, Please define the term persistent protection. It is not clear, what kind of protection author have in mind.

Line 35, the technology of the modification of wood with PF and other resins is rather old. I suggest the authors to elucidate the reasons why this technology can not make major breakthrough like some other modification techniques, like acetylation and thermal modification? Modification of wood with acrylic monomers is not the best comparison. I would pick up the technology at higher TRL, like modification of wood with DMDHEU?

Line 39, I do not think that the better penetration helps us in understanding of the protection phenomena. I do agree that penetration is the key, but I can not see a link between penetration and studying of the penetration phenomena? I think the relationship is pretty clear and is known from wood preservation, good penetration is key for good performance.

Material and methods

In general methodology section is well described. There has been state of the art methodology applied. Methodology is appropriate to address the objectives of the manuscript.

Line 75, sentence needs to be rewritten. Check the grammar.

Line 85, please pay attention on SI rules for units. There should be gap between number and unit.

Regarding the material, authors have to provide more data:

Why respective wood species was selected? Did you work on sapwood or heartwood? What is the portion of each tissue. If sapwood was not distinguished from heartwood, penetration studies are useless and impossible to comment. What are the ringwidth? What is the density? Were there any growth anomalies present? Did you technically dry the wood before processing? What is the climate in respective plantation?

It is not clearly stated if uptake of treatment solution and retention were determined. These are the key data for any kind of interpretation. Please describe methodology more precisely.

Line 120, I am not sure if leica MZ6 is microtome. I recall this device as classic stereomicroscope?

Figure 2, the surface of pine is almost uniform. What is the reason for that? With exception of the resin canals we can not see porous structure? Is micropolishing the best technique for wood surface preparation? How polishing affects the surface?

Authors should provide more data about no of replicates?

Results

Gravimetrical data are missing.

Line 165, if wood samples has been treated with help of pressure, can we talk about diffusion?

Did you notice any deposits on the surface, or potential chromatographic effect?

Figure 4, define outer and inner layer? Is this typical or best image?

There is no statistical analysis performed.

Author Response

Dear Reviewer,

Thank you so much for spending much time to give us lots of good comments on our manuscript and giving us the opportunity to revise our manuscript.

Round 2

Reviewer 1 Report

Revised version of this manuscript still misses some crucial experiments, such as the mechanical properties characterization in terms of hardness, compression strength, shear stress, etc. Or the physical or chemical characterization in terms of odor, colour, water resistance, etc.

FTIR spectra shows the range between 1800-800 cm-1. However, it would be also useful to analyze the range between 3500-3000 cm-1, where the OH- groups appear. Authors described the interaction between wood and PF resin through -OH groups, so FTIR analysis should be carried out in the range between 3500-800 cm-1.

Page 1, line 39: sentence started with "In the past decades.... on the modification efficiency" should be reformulated to be well understood.

Page 2, line 87: 24st is wrong, I think authors mean 24th.

Author Response

Appreciate your good comments on our manuscript. We will improve our work in future. Thanks again!

Reviewer 3 Report

Manuscript has been considerably improved. Authors have address all of the comments from the first round of review. Changes are clearly evident in the manuscript as well as in respective response. I suggest the editors to accept paper in present form. 

Author Response

(The authors gave the same response as above.)
